# Isoindigo–Thiophene D–A–D–Type Conjugated Polymers: Electrosynthesis and Electrochromic Performances

**DOI:** 10.3390/ijms24032219

**Published:** 2023-01-22

**Authors:** Jie Cao, Xiaoyu Luo, Shenglong Zhou, Zhixin Wu, Qi Zhao, Hua Gu, Wen Wang, Zhilin Zhang, Kaiyue Zhang, Kaiyun Li, Jingkun Xu, Ximei Liu, Baoyang Lu, Kaiwen Lin

**Affiliations:** 1Jiangxi Key Laboratory of Flexible Electronics, Flexible Electronics Innovation Institute, Jiangxi Science and Technology Normal University, Nanchang 330013, China; 2School of Chemistry and Chemical Engineering, Jiangxi Science and Technology Normal University, Nanchang 330013, China; 3School of Pharmacy, Jiangxi Science and Technology Normal University, Nanchang 330013, China; 4School of Chemistry and Molecular Engineering, Qingdao University of Science and Technology, Qingdao 266042, China; 5Department of Materials and Food, University of Electronic Science and Technology of China Zhongshan Institute, Zhongshan 528402, China

**Keywords:** isoindigo, thiophene, donor–acceptor–donor polymers, electrochromic, electrosynthesis

## Abstract

Four novel isoindigo–thiophene D–A–D–type precursors are synthesized by Stille coupling and electrosynthesized to yield corresponding hybrid polymers with favorable electrochemical and electrochromic performances. Intrinsic structure–property relationships of precursors and corresponding polymers, including surface morphology, band gaps, electrochemical properties, and electrochromic behaviors, are systematically investigated. The resultant isoindigo–thiophene D–A–D–type polymer combines the merits of isoindigo and polythiophene, including the excellent stability of isoindigo–based polymers and the extraordinary electrochromic stability of polythiophene. The low onset oxidation potential of precursors ranges from 1.10 to 1.15 V vs. Ag/AgCl, contributing to the electrodeposition of high–quality polymer films. Further kinetic studies illustrate that isoindigo–thiophene D–A–D–type polymers possess favorable electrochromic performances, including high optical contrast (53%, 1000 nm), fast switching time (0.8 s), and high coloration efficiency (124 cm^2^ C^−1^). These features of isoindigo–thiophene D–A–D–type conjugated polymers could provide a possibility for rational design and application as electrochromic materials.

## 1. Introduction

Electrochromic (EC) material is an indispensable part of electrochromic devices and has received extraordinary attention in diverse applications, including smart windows [1,2], photochromic lenses [3,4], electronic displays [5,6], and military camouflage [7,8], due to its distinctive polymeric nature, low energy consumption, facile synthesis, and superior coloration efficiency [9]. Considerable efforts have been dedicated to exploring and designing novel EC materials that possess high electrochromic performance. One effective strategy to enhance the electrochromic performance is to employ the donor–acceptor–donor (D–A–D) method [10,11,12,13,14]. Owing to the ordered alternation structure between acceptor units (electron–withdrawing) and donor units (electron–donating) on backbones, D–A–D–type polymers display a strong intramolecular charge transfer feature, endowing EC materials with low band gaps and a tunable electrochromic performance [15]. Conjugated polymers, such as polythiophenes, polypyrroles, polyanilines, etc., have emerged as ideal donor unit candidates due to their multicolor feature, narrow optical absorption region, changeable band gaps, and optical contrasts in designing high–performance EC materials [16]. Several studies demonstrated that various D–A–D–type EC conjugated polymers with thiophene as the donor unit display a lower band gap and high electrochromic performance [13,17,18,19,20,21]. For example, Liu et al. reported three D–A–D–type polymers, including PThQ–Ph, PThQ–PhOMe, and PThQ–T, which comprise thiophene–substituted quinoxaline derivates. The obtained EC materials displayed high optical contrast (over 80%) as well as high coloration efficiency (~300 cm^2^ C^−1^) [13]. Our team electropolymerized a set of chalcogenodiazolo[3,4–c]pyridine–based D–A–D–type EC–conjugated polymers with various thiophenes as donor units, exhibiting tunable colors with fast switching time (0.3~0.6 s), excellent optical contrast (~37%), and favorable redox stability [10]. 

Ever since isoindigo is first used in organic photovoltaics by Reynolds’s group [15,22], isoindigo has received enormous research interest as an acceptor unit [15,23,24,25]. As a geometric isomer of the renowned indigo dye, isoindigo consists of two five–membered lactam rings conjugated by a double bond, and each lactam is fused with a benzene ring to obtain a fully conjugated structure [17,26,27], which renders them with high electron–withdrawing nature, good planarity, and the lowest unoccupied molecular orbital (LUMO) energy level [28,29]. In prior work [17], we prepared a set of isoindigo–based D–A–D–type EC conjugated polymers with EDOT or BisEDOT as electron–donating units, and isoindigo as an electron–deficient unit and achieved reversible color changes with fast switching time, excellent coloration efficiency, and favorable redox stability. Apart from altering donor or acceptor units, modifying donor or acceptor units with alkyl side chains can be regarded as an effective strategy to tailor the electrochromic performance of D–A–D–type conjugated polymers. Introducing alkyl chains on thiophene can effectively avoid the cross–coupling reaction and enhance electron–donation ability, thus, improving the solubility of thiophene chains and enhancing the conjugation effect of D–A–D conjugated precursors. This synergistic effect affects the intramolecular charge transfer rate and structural planarity, further affecting the electrochemical and electrochromic performance of polymers [20,30].

Here, we design four new isoindigo–based D–A–D–type precursors (IDOH–Th, IDOD–Th, IDOH–3HT, and IDOD–3HT) with different alkyl side chains modified by Stille coupling isoindigo with thiophene, successfully electrosynthesized to obtain corresponding polymers. Subsequently, we systematically explore the intrinsic structure–property relationships of isoindigo–thiophene D–A–D–type conjugated polymers with different alkyl side chains, including surface morphology, band gaps, electrochemical properties, and electrochromic behaviors. Kinetic studies reveal that PIDOH–3HT displays favorable electrochromic performance, including fast switching time (0.8 s), high optical contrast (53%, 1000 nm), and high coloration efficiency (124 cm^2^ C^−1^). Combined with our previous work, isoindigo–based D–A–D–type polymers can be rationally designed to improve electrochromic performance, which is suitable for diverse applications, such as electronic displays, smart windows, and flexible electronics.

## 2. Results and Discussion

### 2.1. Synthesis of Isoindigo–Thiophene D–A–D–Type Precursors 

Combining isoindigo with strong electron–withdrawing ability and thiophene would possibly create excellent electrochromic materials. Four isoindigo–thiophene D–A–D–type precursors were designed and synthesized to investigate the electrochromic performance. These precursors are modified by a D–A–D structure comprising isoindigo as the acceptor unit and thiophene (Th)/3–hexylthiophene (3HT) moieties as the donor units. The synthesis routes of target precursors are shown in Figure 1; the final product yields are typically more than 60%. All targeted precursors are confirmed by ^1^H and ^13^C NMR spectra (Appendix A).

### 2.2. Optical Characterization 

The optical properties of precursors are examined by UV–vis absorption spectra in CH_2_Cl_2_. The maximum absorption wavelength (*λ*_max_) and optical band gap (*E_g_* = 1240/*λ*) values are summarized in Table 1. Similar to classic D–A–D–type precursors, all precursors exhibit dual–band absorptions, that is, π–π* transition of conjugated systems at the higher–energy band (340–440 nm) and intramolecular charge transfer between the donor and acceptor unit at the broader low–energy band (450–550 nm) (Figure 1) [18,30]. Compared with IDOH–Th and IDOD–Th, IDOH–3HT and IDOD–3HT display a slightly red–shifted phenomenon, demonstrating that the donor moieties enhance the conjugated degree and change the electronic configuration of precursors [16,31]. Moreover, the absorption maximum for IDOH–3HT is bathochromically shifted, which is ascribed to an increment in the conjugated degree.

### 2.3. Electrochemical Polymerization 

The electrochemical behavior of precursors is examined in CH_2_Cl_2_–Bu_4_NPF_6_ (0.1 mol L^−1^). As shown in anodic polarization curves (Figure 2A,B), the onset oxidation potential (*E_onset_*) of IDOH–3HT (1.10 V) is lower than IDOH–Th (1.13 V), IDOD–Th (1.15 V), and IDOD–3HT (1.15 V). Compared with IDOH–Th, IDOD–Th possesses a relatively high onset oxidation potential, mainly attributed to the longer alkyl chains on the lactam rings in the isoindigo structure. The existence of a long alkyl chain creates a greater steric hindrance and enhances the HOMO level of precursors, leading to a rise in onset oxidation potential and polymerization difficulty [30]. Additionally, introducing hexyl side chains on donor moieties can enhance the electron–donating ability and provide a lower onset oxidation potential to create a favorable condition for the electrosynthesis of the precursors [10]. Meanwhile, IDOHE and IDODE (Appendix A), employing a higher electron–donating unit of EDOT as the donor, exhibit lower onset oxidation potentials than the above isoindigo–thiophene D–A–D–type precursors (Table 1). To investigate the electrochemical properties of precursors, we implement cyclic voltammetry (CV) characterization for all precursors. The distinctive difference in cyclic voltammograms (Figure 2C–F) among precursors can be found during the potential scanning process. The redox peak current densities slowly rise during the repeating CV scan, which manifests the growth of a polymer on the electrode. It is noted that the anodic peak shifts to a positive potential and the cathodic peak shifts to a negative potential during the process of polymer deposition. This phenomenon is mainly attributed to the fact that higher overpotential is needed to conquer the increase in resistance caused by the formation of polymer films [28,32]. We further prepare isoindigo–thiophene D–A–D–type polymers by employing the electrosynthesis method to investigate their properties. The polymerization potentials for IDOH–Th, IDOD–Th, IDOH–3HT, and IDOD–3HT are optimized to be 1.30, 1.35, 1.25, and 1.30 V vs. Ag/AgCl, respectively. At these corresponding applied potentials, all isoindigo–thiophene D–A–D–type polymers are electrosynthesized utilizing the chronoamperometry method in CH_2_Cl_2_–Bu_4_NPF_6_ (0.10 mol L^−1^) with ITO as the working electrode, Ag/AgCl as the reference electrode, and Pt wires as the counter electrode (Appendix A).

### 2.4. Theoretical Calculations

The optimal configuration of precursors is simulated by density functional theory (DFT) at the B3LYP/6–31G* level. Obviously, all precursors have the same electron cloud distribution on the frontier molecular orbital (Figure 3). The electron cloud on the LUMO orbital is mainly localized on the strong electron–withdrawing structure of isoindigo, while the highest occupied molecular orbital (HOMO) is mainly delocalized on the electron–donating structure of thiophene. The HOMO–LUMO gap of IDOH–3HT is smaller than other precursors, signifying that it possesses preferable planar structures. This phenomenon is due to the π–electron coupling decreasing the HOMO–LOMO band gaps of the corresponding polymers, which facilitates precursors to polymerize easily [13,32,35]. IDOH–3HT also has a stronger conjugated degree than other precursors. The calculated LUMO energy level of IDOH–3HT at −2.80 eV is lower than IDOH–Th (–2.85 eV), IDOD–Th (–2.85 eV), and IDOHD–3HT (–2.80 eV), which illustrates that the elongation of the alkyl side chain has a distinct influence on the energy levels.

### 2.5. FT–IR Spectra

FT–IR spectra of the precursors and corresponding polymers are measured to interpret polymer structure and clarify the polymerization mechanism (Appendix A). The detailed assignment of FT–IR spectral absorption peaks is listed in Appendix A. From these results, the peak of IDOH–Th and IDOD–Th at 1110/890 cm^−1^ in the FT–IR spectrum could be assigned to the =C–H in–plane deformation and out–of–plane vibration of the thiophene ring, respectively. However, these characteristic peaks mentioned above are inclined to weaken and disappear, which shows that the electrochemical polymerization of precursors mainly occurs at the α or β position. The absorption peaks at around 3070~3090/3097 cm^−1^ (IDOH/IDOD–Th) and 2841/2870 cm^−1^ (IDOH/IDOD–3HT) of thiophene units (= C–H vibration of thiophene) vanish in the spectra of corresponding polymers, proving that IDOH and IDOD are electropolymerized via coupling at the α positions of thiophene units. The peak of the precursor at 2903~2967 cm^−1^ is assigned to the C–H bonds in the alkyl side chains, and the peak of the corresponding polymer remains essentially unchanged (2838~2957 cm^−1^), confirming that the side alkyl chains on acceptor/donor units are not damaged in the process of electrosynthesis. Take PIDOH–3HT as an example: the peak at 726 cm^−1^ is characteristic of the out–of–plane deformation vibration of =C–H bonds on the benzene ring, while the peaks around 1595~1610 cm^−1^ and 2860~2952 cm^−1^ result from stretching vibration in benzene rings and C–H bonds in the alkyl side chains, respectively. Additionally, the peaks at 846 cm^−1^ in the spectra can be ascribed to the introduction of the dopant ion PF_6_^−^ [27,36,37].

### 2.6. Morphology 

The electrochemistry and electrochromic performance of polymer films are closely related to their surface morphology. To investigate the morphology and microstructure of polymer films, we analyze scanning electron microscope images (SEM) of polymer films, which are directly electrodeposited on ITO glass. At a magnification of ×20,000, PIDOD–Th, PIDOH–3HT, and PIDOD–3HT films with hollow morphology exhibit different surface morphologies under doped and dedoped states (Appendix A). The PIDOH–Th film displays a smooth and compact surface morphology under doped and dedoped states (Appendix A). After dedoping, the morphology of the PIDOH–Th film shows no obvious difference, implying that dedoping processes do not destroy the surface morphologies of the doped PIDOH–Th. For PIDOD–Th, the polymer films are wrinkled both in dedoped and doped states, showing poor electrochemical and electrochromic behaviors (Appendix A). Compared with other polymer films, PIDOH–3HT films exhibit uniform morphologies with hollow structures (Appendix A), which is beneficial to ions passing through the polymer film during the doping/dedoping process. PIDOD–3HT shows a very different morphology in both dedoped and doped states, with different degrees of aggregation (Appendix A), which is unfavorable to obtaining excellent electrochromic performances.

### 2.7. Electrochemistry of Polymer Films 

The electrochemical behavior of polymers is detected at varying scan rates ranging from 300 mV s^−1^ to 25 mV s^−1^ (Figure 4A–D) by utilizing the cyclic voltammetry (CV) test. The CVs of polymer films display relatively broad redox waves and a favorable linear relationship between the potential scan rate and redox peak current densities, which reveals that polymer films are firmly attached to the Pt electrode (Figure 4E–H) [14]. Compared with PIDOD–Th and PIDOD–3HT, PIDOH–Th and PIDOH–3HT exhibit relatively high current density values with better linearity fitting and outstanding electrochemical stability, with less than 9% electroactivity loss, even after 500 cycles (Appendix A), signifying that favorable electrochemical activity can be achieved by introducing alkyl chains of a certain length on the main chain. Meanwhile, the CVs of polymer films display a remarkable hysteresis, where the redox peaks drift at varying sweep rates with the scan rate decreasing due to the competition between the scan rates and the anion migration rates [2,19].

### 2.8. Spectroelectrochemistry

Optical absorption spectra of PIDOH–Th, PIDOD–Th, PIDOH–3HT, and PIDOD–3HT films are explored under different applied voltages in CH_3_CN–Bu_4_NPF_6_ (0.1 mol L^−1^) (Figure 5). All polymers possess two maximum absorption peaks where the absorption bands at the longer wavelengths originate from the π–π* transition of conjugated structures and at the shorter wavelengths can be attributed to the charge transport between donor and acceptor units [15,20,38]. Particularly, the absorption maxima for PIDOH–3HT and PIDOD–3HT bathochromically shift compared with PIDOH–Th and PIDOD–Th, suggesting that the introduction of side alkyl chains on donor units not only increases the electron–donation ability but also improves electrochemical and optical performance. Furthermore, the optical band gaps of polymers exhibit the following trend: 1.91 eV (PIDOH–Th/PIDOD–Th) > 1.89 eV (PIDOD–3HT) > 1.87 eV (PIDOH–3HT). This phenomenon can be explained by PIDOH–3HT films having better conjugated structures with the elongation of alkyl side chain length on donor units. As the applied voltage increases, the peak absorption intensity slowly decreases or even diminishes, and a new optical absorption peak appears in the near–infrared region, which proves the formation of polaron and bipolaron in the process of electrochemical polymerization [16,18,39]. During the transition from the neutral state to the oxidized state (Figure 5A,B), PIDOH–Th and PIDOD–Th films display no clear color changes and remain light gray between the doped and dedoped states. In contrast, PIDOH–3HT and PIDOH–3HT films reveal distinct colors ranging from dark blue in the neutral state to blue gray or gray in the oxidized state. 

### 2.9. Electrochromic Performance 

To further evaluate the electrochromic performance of polymers, the kinetic studies of PIDOH–Th, PIDOD–Th, PIDOH–3HT, and PIDOD–3HT were carried out by using double–step chronoamperometry in the CH_3_CN–Bu_4_NPF_6_ electrolyte solution. The electrochromic parameters of these polymers, such as optical contrast (ΔT), response time, and coloration efficiency (CE), are listed in Table 2. Notably, the optical contrast of PIDOH–3HT is 34.1% at 648 nm and 52.5% at 1000 nm, which is much higher than polythiophene (15% at 465 nm; 9% at 750 nm) [15] and other polymers (PIDOH–Th: 1% at 364 nm, 3% at 604 nm, 6% at 1050 nm; PIDOH–Th: 1.5% at 900 nm; PIDOD–3HT: 2% at 415 nm, 4% at 611 nm,11% at 848 nm) (Figure 6). This phenomenon can be consistent with the loose morphology of PIDOH–3HT (Appendix A), which facilitates the immigration and emigration of dopant ions and greatly enhances the electrochromic properties. The excessive elongation of alkyl side chain length will improve the solubility of polymers, rendering polymers with unstable electrochromic properties. Additionally, coloration efficiency reflects the coloration degree of the material. The time–transmittance curves clearly reveal that PIDOH–3HT possesses the highest coloration efficiency value of 124 cm^2^ C^−1^ at 1000 nm and the fastest switching time (0.8 s, 648 nm), which is superior to PIDOH–Th, PIDOD–Th, and PIDOD–3HT.

## 3. Materials and Methods

### 3.1. Materials

6–Bromooxoindole–2–dione (98%), tetrahydrofuran (AR; THF), acetate (99.5%), 1–iodododecane (AR), and 6–bromoindoline–2,3–dione (98%) were purchased from Shanghai Vita Chemical. Hydrochloric acid (AR; HCl), 1–bromohexane (AR), acetonitrile (99%; ACN), tributyltin chloride (97%), 3–hexylthiophene (AR), and *n*–butyllithium solution (1.6 M in hexanes; *n*–BuLi) were obtained from J&K Scientific. Tetrakis (triphenyphosphine) palladium (99%; Pd(PPh_3_)_4_) and dichloromethane (99%; DCM) were acquired from Energy Chemical and Xilong Chemical, respectively. All other chemicals were of analytical grade and directly used as received.

### 3.2. Synthesis

IDOH–Th, IDOD–Th, IDOH–3HT, and IDOD–3HT were synthesized through Stille coupling. The explicit synthetic routes are described in Figure 1. (E)–6,6′–dibromo–1,1′–dihexyl–[3,3′–biindolinylidene]–2,2′–dione (IDOH), 6,6′dibromoindoledione, and (E)–6,6′–dibromo–1,1′–didodecyl–[3,3′–biindolinylidene]–2,2′–dione (IDOD) were synthesized according to literature procedures [10,13,18].

#### 3.2.1. Tributyl(thiophene–2–yl)stannane and Tributyl(4–hexylthiophene–2–yl)stannane

Thiophene (0.8 mmol) or 3–hexylthiophene (0.8 mmol) was added to the purified THF (20 mL) solution and stirred at –78 °C. During the reaction, *n*–BuLi (13 mmol, 1.6 mol L^−1^ in hexanes) was added by dropping to the system and the reaction lasted for 3 h. When the temperature reached to –48 °C, tributyltin chloride (13 mmol) was added dropwise. Then, the mixture solution was stirred at room temperature for 12 h. Subsequently, the reaction mixture was concentrated under reduced pressure and suitable dichloromethane was added for further purification. The resulting mixture was washed with NH_4_Cl (aq), NaCl (aq), and distilled water until the solution was clear. Finally, the target product was obtained by drying the organic layer with MgSO_4_ and evaporating the solvent.

#### 3.2.2. IDOH–Th and IDOD–Th

IDOH or IDOD (1.0 mmol), tributyl(thiophene–2–yl)stannane (4.0 mmol), and Pd (PPh_3_)_4_ (5.0 mmol) were added into a two–neck flask with degassed toluene (50 mL) and stirred at 110 °C for 24 h under a nitrogen atmosphere. After cooling down to room temperature, the solution was poured into deionized water and extracted with DCM. The residue was purified by column chromatography. IDOH–Th: 64%. ^1^H NMR (400 MHz, CDCl_3_, ppm) δ: 9.19−9.17 (d, J = 8.6 Hz, 4H), 7.29 (s, 4H), 6.95 (s, 4H), 3.84−3.80 (m, 4H), 2.67−2.63 (m, 4H), 1.36−1.26 (s, 12H), 0.90−0.88 (t, J = 6.8 Hz, 6H). ^13^C NMR (400 MHz, CDCl_3_, ppm) δ: 169.1, 145.1, 140.3, 133.7, 128.6, 128.0, 127.6, 127.1, 123.0, 119.4, 42.7, 31.5, 27.4, 26.7, 22.7, 14.1. IDOD–Th: 62%. ^1^H NMR (400 MHz, CDCl_3_) δ 9.17–9.15 (d, J = 8.6 Hz, 4H), 7.28 (s, 4H), 6.96 (s, 4H), 3.84–3.81 (m, 4H), 2.66–2.62 (m, 4H), 1.74–1.65 (m, 8H), 1.87–1.85 (m, 20H), 0.92–0.88 (t, J = 6.8 Hz, 6H). ^13^C NMR (101 MHz, CDCl_3_) δ 168.21, 154.30, 143.91, 137.97, 132.20, 130.46, 128.31, 126.10, 124.36, 121.15, 119.42, 104.93, 40.13, 31.93, 29.71, 29.64, 29.53, 29.34, 27.58, 27.07, 22.69, 14.10.

#### 3.2.3. IDOH–3HT and IDOD–3HT 

IDOH or IDOD (1.0 mmol), tributyl(4–hexylthiophene–2–yl)stannane (4.0 mmol), and Pd (PPh_3_)_4_ (5.0 mmol) were added to degassed toluene (50 mL) and stirred at 110 °C for 24 h under a nitrogen atmosphere. After cooling down to room temperature, the solution was poured into deionized water and extracted with CH_2_Cl_2_, dried with MgSO_4_, and purified by column chromatography. IDOH–3HT: 75%. ^1^H NMR (400 MHz, CDCl_3_, ppm) δ 9.20–9.17 (d, J = 8.4 Hz, 2H), 7.45–7.44 (d, J = 3.3 Hz, 2H), 7.37–7.36 (d, J = 5.0 Hz, 2H), 7.14–7.13 (d, J = 8.4 Hz, 2H), 6.99 (d, J = 4.8 Hz, 2H), 3.85 (t, J = 7.2 Hz, 4H), 1.76–1.72 (m, 4H), 1.35–1.27 (m, 32H), 0.88 (t, J = 6.5 Hz, 12H). ^13^C NMR (400 MHz, CDCl_3_, ppm) δ: 167.82, 144.80, 144.17, 143.14, 137.67, 131.45, 129.80, 125.19, 120.41, 118.71, 104.14, 39.55, 31.15, 30.98, 30.10, 29.92, 28.49, 27.02, 26.19, 22.09, 22.02. IDOD–3HT: 64%. ^1^H NMR (400 MHz, CDCl_3_) δ 9.22–9.20 (d, J = 8.4 Hz, 2H), 7.29–7.28 (d, J = 3.3 Hz, 2H), 7.14–7.12 (d, J = 5.0 Hz, 2H), 7.02 (s, 2H), 7.01 (s, 2H), 3.82–3.79 (t, J = 7.2 Hz, 4H), 2.74–2.72 (m, 4H), 1.73–1.65 (m, 10H), 0.88–0.83 (m, 36H). ^13^C NMR (101 MHz, CDCl_3_) δ 169.10, 145.13, 138.27, 138.30, 133.15, 125.50, 122.64, 121.73, 119.42, 42.74, 31.94, 31.22, 29.66, 29.39, 27.00, 22.72, 22.70, 14.11.

#### 3.2.4. Characterizations

Precursor structures were measured by ^1^H and ^13^C NMR spectra (Bruke A 400 NMR spectrometer, Germany) with chloroform–*d* (CDCl_3_) as the solvent and tetramethylsilane (TMS) as an internal standard. Frontier orbitals and geometric orbital distributions of precursors were systematically investigated by density functional theory (DFT). DFT calculations were conducted with Gaussian 09 at the B3LYP/6–31 + G (d,p) level. Fourier–transform infrared spectroscopy (FT–IR) was performed using a Bruker Vertex 70 Fourier transform infrared spectrometer (Germany). The electropolymerization of four precursors and electrochemical tests were carried out in an electrochemical workstation (EG&G Princeton Applied Research, America). Scanning electron microscopy (SEM) images were obtained from a scanning electron microscope (VEGA II–LSU, Tescan, Czech). Characterizations of the polymers, including spectroelectrochemical and electrochromic kinetic studies, were performed in precursor–free ACN–Bu_4_NPF_6_ (0.1 mol L^−1^) by combining an electrochemical workstation with a UV–vis spectrophotometer (Specord 200 Plus, Germany).

## 4. Conclusions

Four isoindigo–based D–A–D alternating precursors (IDOH–Th, IDOD–Th, IDOH–3HT, and IDOD–3HT) containing thiophene (Th)/3–hexylthiophene (3HT) are designed and synthesized. Due to the substituted structure modification, the corresponding polymers exhibit tunable electrochromic performances and different color changes. With the elongation of the alkyl side chain on donor units, the resulting IDOH–3HT possesses lower onset oxidation potential, offering low–potential electropolymerization to achieve high–quality films. The PIDOH–3HT films exhibit a reversible color change from blue in the dedoped state to gray in the doped state with favorable optical contrast, fast response times, and high coloration efficiency. Powered by these advanced performances, isoindigo–thiophene D–A–D–type conjugated polymers could be prospective materials for electrochromic applications.

## Data Availability

The data are available in this publication and Appendix A.

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
