# Peer review of "Isoindigo–Thiophene D–A–D–Type Conjugated Polymers: Electrosynthesis and Electrochromic Performances"

_ijms, 2023, doi:10.3390/ijms24032219_

Round 1

Reviewer 1 Report (Previous Reviewer 1)

Lu, Lin et al. developed an efficient strategy to synthesize four isoindigo-based D-A-D alternating polymers (PIDOH-Th, PIDOD-Th, PIDOH-3HT, and PIDOD-3HT) containing thiophene (Th)/3-hexylthiophene. Some analytical methods, which include CV, theoretical calculations, FT-IR spectra, morphological characterization, spectroelectrochemistry and electrochromic performance, were carried out to characterize and systematically compare the properties of several substances. Based on the experimental results and speculation, the property differences of different compounds are reasonably explained. Therefore, I propose this work can be published after minor revision as following:

1) In Scheme 1, line 99. The synthesis step in the second step is wrong, the conditions are inconsistent with the product results.

2) Line 118. In consideration of the almost identical electron-donating ability of C6H12- group and C12H25- group, the explanation for comparison of oxidation potential is unreasonable. Authors should give a convincing explanation for the difference of four molecules.

3) Table 1, line 137. The structures of EPTE and EBTE should be provided so that the reader can easily compare their nature based on the comparison of structural differences.

4) Line 285. In order to make it easier for the reader to repeat the experiment, the polarity of the eluent needs to be provided.

5) Line 291-313. Characterization data of IDOH-Th, IDOD-Th, IDOH-3HT and IDOD-3HT should be carefully checked. For example, in 1H-NMR part, the H numbers of these compounds are inconsistent with the structure of compounds.

Author Response

Lu, Lin et al. developed an efficient strategy to synthesize four isoindigo-based D-A-D alternating polymers (PIDOH-Th, PIDOD-Th, PIDOH-3HT, and PIDOD-3HT) containing thiophene (Th)/3-hexylthiophene. Some analytical methods, which include CV, theoretical calculations, FT-IR spectra, morphological characterization, spectroelectrochemistry and electrochromic performance, were carried out to characterize and systematically compare the properties of several substances. Based on the experimental results and speculation, the property differences of different compounds are reasonably explained. Therefore, I propose this work can be published after minor revision as following:

RESPONSE: We greatly appreciate your recognition of the interesting findings in our work, and also thanks so much for your insightful and constructive comments. We have made corresponding revisions according to your detailed comments. Please see the point-by-point response attached below.

1) In Scheme 1, line 99. The synthesis step in the second step is wrong, the conditions are inconsistent with the product results.

RESPONSE: We are truly grateful to the reviewer’s meticulous check. We sincerely apologize for the mistake in the synthetic routes of isoindigo-thiophene D-A-D type precursor and we have carefully revised the error. To clarify this point, we have revised the synthesis route as follows:

Scheme 1. Synthetic routes of isoindigo-thiophene D-A-D type precursors.

2) Line 118. In consideration of the almost identical electron-donating ability of C6H12- group and C12H25- group, the explanation for comparison of oxidation potential is unreasonable. Authors should give a convincing explanation for the difference of four molecules.

RESPONSE: Thank you for giving us constructive suggestions.

On Lines 117-124: “Compared with IDOH-Th, IDOD-Th possess a relatively high onset oxidation potential, mainly attributed to the longer alkyl chains on the lactam rings in the isoindigo structure. The existence of long alkyl chain creates a greater steric hindrance and enhances the HOMO level of precursors, leading to a rise in onset oxidation potential and polymerization difficulty [30]. Additionally, introducing hexyl side chains on donor moieties can enhance the ability of electron-donating and provide a lower onset oxidation potential to create a favorable condition for the electrosynthesis of the precursors [10].

3) Table 1, line 137. The structures of EPTE and EBTE should be provided so that the reader can easily compare their nature based on the comparison of structural differences.

RESPONSE: Thank you for your insightful comment. We have added chemical structures of IDOH-Th, IDOD-Th, IDOH-3HT, IDOD-3HT, IDOHE, IDODE, EPTE, and EBTE in Supplementary Figure S10 (also listed below) to help the reader better compare the electrochemical and optical properties.

Figure S10. Chemical structures of different D-A-D type precursors.

4) Line 285. In order to make it easier for the reader to repeat the experiment, the polarity of the eluent needs to be provided.

RESPONSE: Thank you for giving us constructive suggestions. We completely agree with the reviewer’s view and are terribly sorry to omit related experimental details.

On Lines 293-297: Subsequently, the reaction mixture was concentrated under reduced pressure and suitable dichloromethane (polarity: 3.4) was added to further purification. The resulting mixture was washed with NH4Cl (aq), NaCl (aq), and distilled water (polarity: 10.2) until it was clear. Finally, the target product was obtained by drying the organic layer with MgSO4 and evaporating the solvent.

5) Line 291-313. Characterization data of IDOH-Th, IDOD-Th, IDOH-3HT, and IDOD-3HT should be carefully checked. For example, in 1H-NMR part, the H numbers of these compounds are inconsistent with the structure of compounds.

RESPONSE: Many thanks to the reviewers for your meticulous check. We deeply apologize for the mistake, retest the 1H NMR of some compounds and modify the manuscript.

Reviewer 2 Report (New Reviewer)

The researcher Jie Cao and co-workers have a report entitled “Isoindigo-thiophene D-A-D type conjugated polymers: electro-2 synthesis and electrochromic performances”. This work reports the synthesizing and characterization of a series of isoindigo thiophene-based small molecule derivatives and the corresponding four isoindigo-thiophene-polymer derivatives.

The newly synthesized isoindigo derivied polymers structural, surface morphology as well as electrochemical and electroromics behaviours.  These polymeric materials shows good electrochromic stabilities.

The research article is well-presented and clearly written with appropriate literature, experimental characterization, and analysis data. Therefore, this work has the potential to be published in the International Journal of Molecular Science, so I accept the present article for publication.   

Author Response

The researcher Jie Cao and co-workers have a report entitled “Isoindigo-thiophene D-A-D type conjugated polymers: electro-2 synthesis and electrochromic performances”. This work reports the synthesizing and characterization of a series of isoindigo thiophene-based small molecule derivatives and the corresponding four isoindigo-thiophene-polymer derivatives.

The newly synthesized isoindigo derivied polymers structural, surface morphology as well as electrochemical and electroromics behaviours. These polymeric materials show good electrochromic stabilities.

The research article is well-presented and clearly written with appropriate literature, experimental characterization, and analysis data. Therefore, this work has the potential to be published in the International Journal of Molecular Science, so I accept the present article for publication.

RESPONSE: Many thanks for the reviewer’s kind comments on our manuscript. It is our great pleasure to be favorably commented by such a world-renowned expert in the field of electrochromism.

Reviewer 3 Report (New Reviewer)

The authors have designed and synthesized four isoindigo-thiophene based D-A-D conjugated polymers for advanced electrochromic applications. They have thoroughly characterized the structures, gemometries, morphologies, as well as various kinds of electrochemical properties of the conjugated oligomers and polymers. Generally, the manuscript is presented in a very logical manner and of interest to the broad polymer community. I would recommend it for publication in International Journal of Molecular Sciences after addressing the following concerns:

1. How to synthesize the isoindigo-thiophene D-A-D polymers from the precursors? What equipment and process parameters (voltage, concentration, etc.) are needed for the electrochemical polymerization? The authors may want to add such critical process information to ensure people can repeat the experiments if they want to.

2.  What is the thickness of the conjugated polymer films? How does the thickness affect the degree of conjugatation, solubility, and electronic performance?

3. There are many paragraphs and/or figure captions written in red color throughout the manuscript and supporting information. Unless the authors really want to emphasize the points, they need to clean up the formats and colors according to the Author Guidelines.

4. In Figure 3 (page 6), two "IDOH-3HT" precursors showed up in the same graph, which was very confusing. The authors should double check the compounds and their corresponding geometries.

5. The authors would want to check the spelling and grammar usage all through the manuscript for better readership. To list a few as follows:

a). On Line 288 (Page 11), "were added into a two-neck flash". Does the authors mean "two-neck flask"?

b). On Line 331 (Page 12), "are designed and electrosynthesis". The "electrosynthesis" is not appropriate here.

Author Response

The authors have designed and synthesized four isoindigo-thiophene based D-A-D conjugated polymers for advanced electrochromic applications. They have thoroughly characterized the structures, gemometries, morphologies, as well as various kinds of electrochemical properties of the conjugated oligomers and polymers. Generally, the manuscript is presented in a very logical manner and of interest to the broad polymer community. I would recommend it for publication in International Journal of Molecular Sciences after addressing the following concerns.

RESPONSE: We would like to express our sincere gratitude to the reviewer for your recognition and kind comments on our work. Please find below the point-by-point response to your comments.

  1. How to synthesize the isoindigo-thiophene D-A-D polymers from the precursors? What equipment and process parameters (voltage, concentration, etc.) are needed for the electrochemical polymerization? The authors may want to add such critical process information to ensure people can repeat the experiments if they want to.

RESPONSE: Thank you for giving us constructive suggestions. We completely agree with the reviewer’s view and are sorry to omit related experimental details.

On Lines 134-139: “We further prepare isoindigo-thiophene D-A-D type polymers by employing electrosynthesis method to investigate their properties. The polymerization potentials for IDOH-Th, IDOD-Th, IDOH-3HT, and IDOD-3HT are optimized to be 1.30, 1.35, 1.25, and 1.30 V vs. Ag/AgCl, respectively. At these corresponding applied potentials, all isoindigo-thiophene D-A-D type polymers are electrosynthesized utilizing the chronoamperometry method in CH2Cl2-Bu4NPF6 (0.10 mol L-1), in which ITO as the working electrode, Ag/AgCl as the reference electrode, and Pt wires as the counter electrode (Figure S11).”

Figure S11. Schematic diagram of the equipment based on one-chamber three-electrode system.

  1. What is the thickness of the conjugated polymer films? How does the thickness affect the degree of conjugatation, solubility, and electronic performance?

RESPONSE: Thank you for your insightful and constructive comments. The polymers were prepared by the chronoamperometry method with a charge of about 5 mC. In this electrochemical deposition condition, the thickness of polymer films was about 200 nm, which is in agreement with the reported structure [R1]. The thickness effect of conjugation, solubility, and electronic performance would be studied in another work in detail.

[R1] Hu, B.; Li, CY.; Liu, ZC.; Zhang, XL.; Luo, W.; Jin, L. Synthesis and multi-electrochromic properties of asymmetric structure polymers based on carbazole-EDOT and 2,5-edithienylpyrrole derivatives. Electrochim. Acta 2019, 305, 1-10.

  1. There are many paragraphs and/or figure captions written in red color throughout the manuscript and supporting information. Unless the authors really want to emphasize the points, they need to clean up the formats and colors according to the Author Guidelines.

RESPONSE: Thanks to the reviewer for the question. We have already cleaned up the previous format and color, and the latest revisions are marked in red.

  1. In Figure 3 (page 6), two "IDOH-3HT" precursors showed up in the same graph, which was very confusing. The authors should double check the compounds and their corresponding geometries.

RESPONSE: Thanks a lot for pointing out our defects. we have revised this error as follows:

Figure 3. Optimized geometries and HOMO-LUMO orbital distribution of IDOH-Th (A), IDOD-Th (B) IDOH-3HT (C), and IDOD-3HT (D).

  1. The authors would want to check the spelling and grammar usage all through the manuscript for better readership. To list a few as follows:

a). On Line 288 (Page 11), "were added into a two-neck flash". Does the authors mean "two-neck flask"?

b). On Line 331 (Page 12), "are designed and electrosynthesis". The "electrosynthesis" is not appropriate here.

RESPONSE: We are truly grateful to the reviewer’s meticulous check. We sincerely apologize for the g spelling and grammar in the manuscript. We have carefully read the manuscript and revised these errors with help from a native English speaker. Please check the revised manuscript for detailed modifications, which are already highlighted in red.

On Lines 299-300: “IDOH or IDOD (1.0 mmol), tributyl(thiophene-2-yl)stannane (4.0 mmol), and Pd (PPh3)4 (5.0 mmol) were added into a two-neck flask with degassed toluene.

On Lines 342-344: “Four isoindigo-based D-A-D alternating precursors (IDOH-Th, IDOD-Th, IDOH-3HT, and IDOD-3HT) containing thiophene (Th)/3-hexylthiophene (3HT) are designed and synthesized.

Reviewer 4 Report (New Reviewer)

1. Scheme1 for the synthesis route is not clear especially for the alkyl substituted precursors.  Provide the clear pathway for the synthesis and description of the precursors synthesized

2.  It is mentioned that the onset potential for IDOH-3HT is lower compared to other precursors. However, the difference is only 0.02V.  Is this significant? what is the error in the measurements? 

3.  Reference number and article do not match.  Especially Line# 16 ( reference 17) .  Check all the references

This article in content is worthy of publication in IJMS after appropriate revisions are made according to the comments above

Author Response

  1. Scheme 1 for the synthesis route is not clear especially for the alkyl substituted precursors. Provide the clear pathway for the synthesis and description of the precursors synthesized.

RESPONSE: Thank you for your scrupulous check and constructive comment. To clarify this point, we have revised the synthesis route as follows:

Scheme 1. Synthetic routes of isoindigo-thiophene D-A-D type precursors.

  1. It is mentioned that the onset potential for IDOH-3HT is lower compared to other precursors. However, the difference is only 0.02V. Is this significant? what is the error in the measurements?

RESPONSE: Thank you for your insightful and constructive comments.

The experimental results are obtained by repeating the experiment several times, and the main error of the experimental results originates from the definition of the tangent line. The experimental results can be explained as below:

On Lines 117-124: “Compared with IDOH-Th, IDOD-Th possess a relatively high onset oxidation potential, mainly attributed to the longer alkyl chains on the lactam rings in the isoindigo structure. The existence of long alkyl chain creates a greater steric hindrance and enhances the HOMO level of precursors, leading to a rise in onset oxidation potential and polymerization difficulty [30]. Additionally, introducing hexyl side chains on donor moieties can enhance the ability of electron-donating and provide a lower onset oxidation potential to create a favorable condition for the electrosynthesis of the precursors [10].

  1. Reference number and article do not match. Especially Line# 16 (reference 17). Check all the references.

RESPONSE: We are truly grateful to the reviewer’s meticulous check. We have made modifications in reference according to the reviewer’s comments. Please check the latest reference in the revised manuscript.

This article in content is worthy of publication in IJMS after appropriate revisions are made according to the comments above.

This manuscript is a resubmission of an earlier submission. The following is a list of the peer review reports and author responses from that submission.

Round 1

Reviewer 1 Report

In this manuscript, Lu and coworkers designed and synthesized four isoindigo-thiophene D-A-D type precursors including IDOH-Th, IDOD-Th, IDOH-3HT and IDOD-3HT. Subsequently, the polymers of these precursors (PIDOH-Th, PIDOD-Th, PIDOH-299 3HT and PIDOD-3HT) were achieved by electrochemical method and their electrochromic properties were explored. The assays were well designed and this manuscript is proper organized. Therefore, I recommend its publication after minor revision. The following issues should be considered carefully.

1. The authors should label the name of the sample in each figure for easier comparison.

2. The SEM images of polymers with doping and de-doping are unclear to compare the different.

3. The Redox stability of the polymers affects the performance of the electrochromic device, and the authors should provide this data.

4. The manuscript should be carefully checked again. Some minor mistakes are still present.

5. Some related nice work and reviews should be cited. Angew.Chem. Int. Ed. 2020, 59,1158311590; Chinese Chemical Letters 2022 DOI:10.1016/j.cclet.2022.08.015; Org. Lett. 2022, 24, 19, 34713476;

Reviewer 2 Report

This article appears to be a routine piece of work rather than a significant scientific contribution. Also, changing only the alkyl chain does not work to tune the opto-electronic properties of the designed Isoindo derivatives, as shown in Figure 1. Moreover, many studies with more information have already been done and published (e.g. Chem. Mater. Reynolds et.al., 2014, 26, 1, 664–678: https://doi.org/10.1021/cm402219v; Tetrahedron Letters., 2016, 57, 5856–5858: https://doi.org/10.1016/j.tetlet.2016.11.053). The purpose of this study isn't clearly explained in the introduction. If the electrochemical synthesis process is the most interesting part of this article, it should be made clear in the beginning, along with details about how to characterise polymer films, including their electronic and opto-electrical properties.